# Contralateral Snare Cannulation vs. Retrograde Gate Cannulation during Endovascular Aortic Repair in Difficult Iliac Artery Anatomy: A Single Center Experience

**DOI:** 10.3390/jcm13010175

**Published:** 2023-12-28

**Authors:** Giuseppe Sena, Rossella Montemurro, Francesco Pezzo, Rosario Gioffrè, Giuseppe Gallelli, Paolo Rubino

**Affiliations:** Department of Vascular Surgery, “Pugliese-Ciaccio” Hospital, 88100 Catanzaro, Italy; montemurro.rossella@libero.it (R.M.); fpezzo@libero.it (F.P.); rosario.gioffre@libero.it (R.G.); giuseppegallelli@hotmail.it (G.G.); paolorubinocz@virgilio.it (P.R.)

**Keywords:** abdominal aortic aneurysm, endovascular aneurysm repair, iliac tortuosity index

## Abstract

Objective: Endovascular aneurysm repair is well established as the gold standard in treating abdominal aortic aneurysms. Generally, endovascular repair is performed using a bi or trimodular stent graft, requiring placement of a contralateral iliac limb. Deployment of the contralateral iliac limb requires retrograde gate cannulation of the endograft main body contralateral limb. This step represents the crucial point of a standard endovascular repair procedure and can become challenging, especially in the case of high iliac tortuosity. This study compares the procedural times between the retrograde gate cannulation and the contralateral snare cannulation to demonstrate the possibility of directly performing the contralateral snare cannulation in the case of a complex iliac anatomy assessed by the iliac tortuosity index. Methods: One hundred and forty-eight patients with infrarenal abdominal aortic aneurysms who underwent endovascular aneurysm repair from 2017 to 2022 were analyzed retrospectively. Cannulation times between retrograde gate cannulation and contralateral snare cannulation were compared for each degree of iliac tortuosity. The degree of iliac tortuosity was assessed through the iliac tortuosity index. Cannulation times were detected from inserting the wire into the introducer to passing through the radio-opaque gate markers. Results: The cannulation times were 2.94 min for the retrograde gate cannulation group and 3.15 min for the contralateral snare cannulation group, respectively, with no statistically significant differences (*p* = 0.33). Overall cannulation times were 2.98 min. For the iliac tortuosity index grade 0, the cannulation times were 2.71 min for the retrograde gate cannulation group and 3.85 min for the contralateral snare cannulation group, respectively, with a significant difference in favor of the retrograde gate cannulation group (*p* < 0.0001). For the iliac tortuosity index grade 1, the cannulation times were 2.74 min for the retrograde gate cannulation group and 2.8 min for the contralateral snare cannulation group, respectively, with no statistically significant differences (*p* = 0.63). Regarding the iliac tortuosity index grades 2 and 3, the cannulation times were 3.01 and 4.93 min for the retrograde gate cannulation group and 2.71 and 3.28 min for the contralateral snare cannulation group, respectively. The first group’s times were significantly higher than the second group’s (*p* = 0.01 and *p* = 0.0001). Conclusions: In patients with infrarenal abdominal aortic aneurysms undergoing endovascular aortic repair, the gate cannulation times were significantly shorter for the contralateral snare cannulation method than the retrograde gate cannulation method in the iliac tortuosity index grades 2 and 3. Therefore, performing the contralateral snare cannulation method would be appropriate.

## 1. Introduction

Endovascular aneurysm repair (EVAR) is well established as the gold standard in the treatment of abdominal aortic aneurysms (AAA) [1,2,3]. Indeed, technological advancements in stent-graft design and deployment techniques have led to better outcomes and durability. Generally, EVAR is performed using a bi or trimodular stent graft, requiring a contralateral iliac limb (CL) placement. Deployment of a CL requires retrograde gate cannulation (RGC) of the endograft main body contralateral limb using different combinations of wires and catheters from the contralateral access site to select the gate. This step represents the crucial point of the standard EVAR procedure and can become challenging, especially in the case of high iliac tortuosity [4]. The difficulty of retrograde cannulation increases fluoroscopy times, complication rates, material consumption, and costs. Consequently, several rescue techniques have been developed to overcome these challenges. In particular, one of these is the contralateral snare cannulation (CSC) or cross-over technique, which consists of the wire introduction through the ipsilateral iliac limb after the main prosthetic body deployment and its recovery using a snare device from the endograft main body contralateral limb gate [5,6]. This study aims to compare the procedural times between the RGC and the CSC to demonstrate the possibility of directly performing the CSC in the case of complex iliac anatomy assessed by the iliac tortuosity index (ITI).

## 2. Methods

This retrospective, case–control, single-center study analyzed 148 patients with infrarenal AAA who consecutively underwent EVAR between 2017 and 2022. The study obtained institutional ethics committee approval and only included elective EVAR patients, excluding those who underwent emergency EVAR. The study used various commercially available modular stent grafts, including Endurant II (METRONIC INC., Dublin, Ireland), Treo (BOLTON MEDICAL INC., Sunrise, FL, USA), and Zenith (COOK MEDICAL INC., Bloomington, IN, USA) chosen based on availability and operator preference. All cases were performed in a hybrid room with fixed rotational angiography. The primary endpoint was to evaluate differences in gate cannulation times between the RGC and CSC methods during the EVAR procedure for the tortuosity of the iliac arteries. Iliac tortuosity was assessed using the ITI, defined as the ratio between the center lumen line distance between the common femoral artery and the aortic bifurcation and the straight-line distance between the common femoral artery and the aortic bifurcation [7]. Iliac tortuosity was classified into four degrees (ITI grades are shown in the Table 1). Cannulation times between the CSC and RGC were compared for each degree of iliac tortuosity. Times were measured from inserting the wire into the introducer to passing through the radio-opaque gate markers. The RGC method was attempted for at least 15 min, and if unsuccessful, the CSC technique was followed. Patients switched to the CSC were counted only in the CSC group. The first operator performed the cannulation check, which was taken into consideration during time measurement. Patient demographics, total operating times, anatomical features of aneurysms, type of stent graft, type of equipment, and adverse procedure events were collected. Statistical analyses were conducted, continuous variables were reported by means and standard deviations, and categorical variables were reported as frequencies and percentages. Chi-square, Fisher exact tests, and *T*-tests were used to evaluate the differences between the variables. A *T*-test was used to assess the difference in cannulation times between the CSC and RGC, and corresponding 95% confidence intervals (CI) were estimated. A statistically significant difference was considered when *p* < 0.05. Statistical analysis was performed in November 2022 using SPSS (version 25.0) software (IBM Corp., Armonk, NY, USA).

### Surgical Technique

During the RGC procedure, the common femoral artery was surgically exposed and punctured with a Seldinger needle. A 6F sheath was then inserted. Afterward, a 0.035-inch Terumo floppy guidewire (Bolton Medical Inc., Sunrise, FL, USA) was advanced, followed by a 5F Cobra catheter (Bolton Medical Inc., Sunrise, FL, USA). The 0.035-inch guide wire was replaced with a Lunderquist extra stiff wire (Cookmedical Inc., Bloomington, IN, USA). A tail catheter was inserted through the contralateral access and advanced on the 0.035-inch guidewire. Once the 6F sheath was removed, the main body of the endograft was advanced and released. Cannulation of the contralateral gate was then performed using a 0.035-inch guidewire (as shown in Figure 1). The extra stiff guide was positioned, and the contralateral limb was deployed (as shown in Figure 2). The ipsilateral limb was then advanced and deployed, and finally, the femoral artery and wound were closed.

In the CSC procedure, after the endograft’s main body was released, a 0.035-inch guidewire was passed through the contralateral gate (as shown in Figure 3). Curved catheters, such as the Simmons (MED-ITALIA BIOMEDICA SRL, Genova, Italia), were used to facilitate this operation. The guidewire was then captured using an Amplatz Goose Neck (Cook Medical Inc., Bloomington, IN, USA) catheter and advanced from the contralateral access (as shown in Figure 4). The Cobra 5F catheter was advanced on the guidewire, and the 0.035-inch guidewire was replaced with a Lunderquist extra stiff guidewire. The contralateral limb was then advanced and deployed, followed by the ipsilateral limb.

## 3. Results

Between 2017 and 2022, a total of 148 patients were enrolled in a study—out of these, 116 patients underwent RGC, and 32 underwent CSC. The patients’ mean age was 73.4 years, and 54% were male. The two groups were similar in age, gender, aneurysmal neck length, aortic diameter at the gate, aortic lumen at the gate, and the type of stent graft used. However, there was a significant difference between the two groups for iliac tortuosity grades 0 and 3 but not for grades 1 and 2. More patients were in the retrograde group, with an ITI grade 0, and more in the snare group, with an ITI grade 3. In addition, the distance between the aortic bifurcation and the gate was significantly greater in the snare group than in the retrograde group (Table 2). The cannulation times for the RGC and CSC groups were 2.94 min and 3.15 min, respectively, with no significant differences (*p* = 0.33). The overall cannulation time was 2.98 min. However, significant differences were observed in terms of ITI. For the ITI grade 0, the RGC group’s cannulation time was 2.71 min, and the CSC group’s time was 3.85 min, with a significant difference in favor of the RGC group (*p* < 0.0001). For the ITI grade 1, the cannulation times were 2.74 min for the RGC group and 2.8 min for the CSC group, respectively, with no significant differences (*p* = 0.63). For the ITI grades 2 and 3, the cannulation times were 3.01 and 4.93 min for the RGC group and 2.71 and 3.28 min for the CSC group, respectively. The RGC group’s times for the two grades were significantly higher than the CSC group’s times (*p* = 0.01 and *p* = 0.0001) (Table 3).

## 4. Discussion

The first-ever EVAR procedure was performed by Parodi et al. in 1991 [8]. Since then, there have been significant improvements in stent-graft materials, designs, and delivery techniques, leading to better outcomes and increased reliability. As a result, EVAR has become the preferred treatment for infrarenal abdominal aortic aneurysms. However, several factors can impact the success of the endovascular procedure, including patient selection, optimal planning/sizing, and attention to detail during the procedure’s execution. Most stent grafts currently available have a trimodular design that requires cannulation of the main body’s contralateral short limb, making gate cannulation and guidewire positioning critical and challenging points of the procedure. Difficulties with gate cannulation are common, particularly with giant aneurysms and high iliac tortuosity [9]. Starting with the snare method can directly optimize the entire procedure by reducing operating time, material consumption, and radiation dose and avoiding repeated attempts before switching. In cases of difficult gate cannulations, several options are available, such as the crossed limb technique, which connects the ipsilateral guidewire to the contralateral gate by deploying the graft limb like a ballerina [10]. However, the postoperative outcomes of this technique have yet to be clarified, and no indications or contraindications have been described. A recent study has shown that this configuration supported high wall shear stress and helicity characteristics [11,12]. Alternatively, brachial access can be performed, but this approach is associated with several complications, such as bleeding and thrombosis [13]. Several factors may influence the success of gate cannulation, including surgeon experience and endograft design. Some studies have compared the two techniques, but the effectiveness of various methods has yet to be evaluated. Titus et al. demonstrated no difference in mean cannulation times between the two methods and that if gate cannulation is not achieved by the retrograde technique in the first 5 min, a crossover to snare is more effective [6]. Others have tried to identify methods that could simplify the procedure, such as the position of the gate on deployment or an increase in its size [14,15]. However, the effectiveness of these methods has yet to be evaluated. Based on our experience with 148 patients who underwent EVAR in the last five years, we compared the cannulation times between the RGC and CGC for each degree of iliac tortuosity according to ITI. The two groups were relatively homogenous, except for the significantly higher number of patients in the retrograde group for ITI grade 0, the significantly higher number of patients in the snare group for ITI grade 3, and the distance between the aortic bifurcation and the gate, which was notably more significant in the snare group than in the retrograde group. Although there was no significant difference regarding the overall cannulation times between the two approaches, in the case of ITI grades 3 and 4, the cannulation times using the snare method were significantly reduced compared to the retrograde method. This result demonstrates that performing a CSC directly compared to RGC, with a potential reduction in overall operating times, radiation dose, amount of materials consumed, and costs, is rational in the case of high iliac tortuosity. Our study is the first to compare gate cannulation times between the RGC and CSC for each degree of iliac tortuosity assessed with ITI. However, this study has several limitations, including its retrospective nature, single-center design, small sample size, and heterogeneity of the two groups for some parameters.

## 5. Conclusions

In patients with infrarenal AAA undergoing EVAR, the gate cannulation times were significantly shorter for the CSC method than for the RGC method in the ITI grades 2 and 3. Therefore, in these cases, it would be appropriate to perform the CSC method directly. Given the study’s limitations, however, further studies, especially randomized and multicenter trials, are needed in the future.

## Figures and Tables

**Figure 1 jcm-13-00175-f001:**
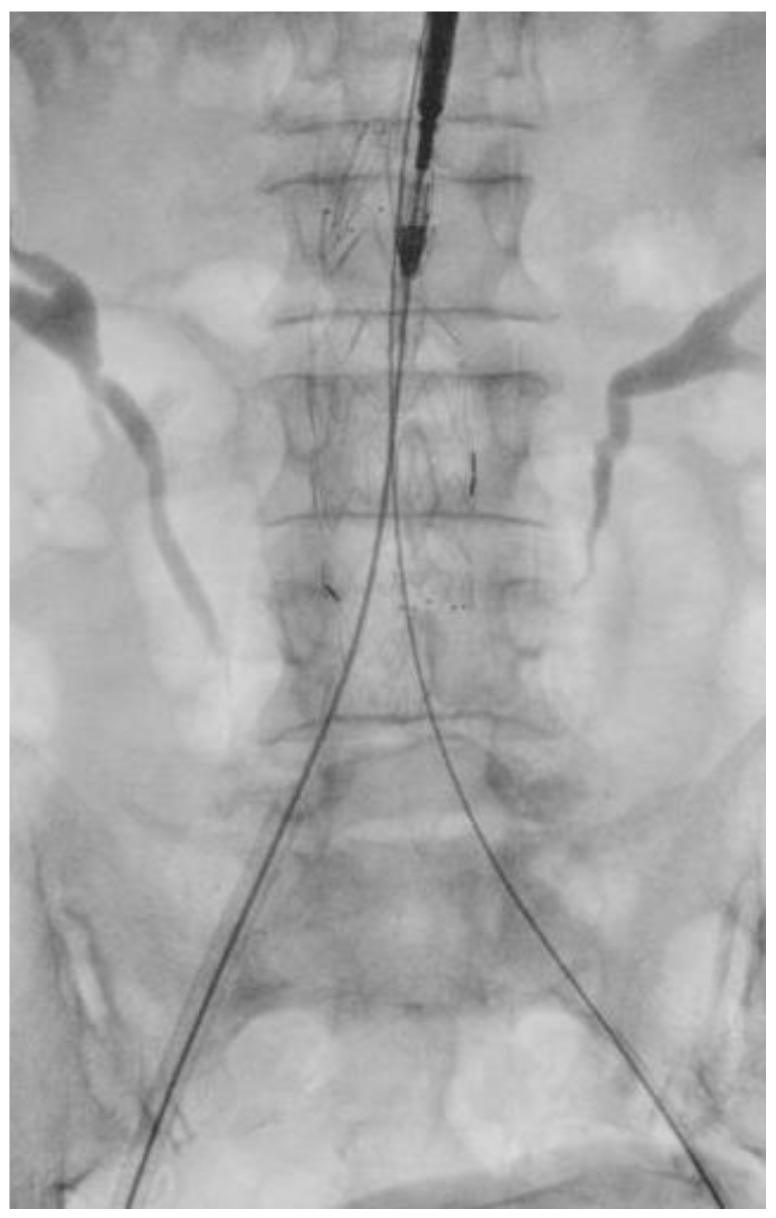
Retrograde gate cannulation.

**Figure 2 jcm-13-00175-f002:**
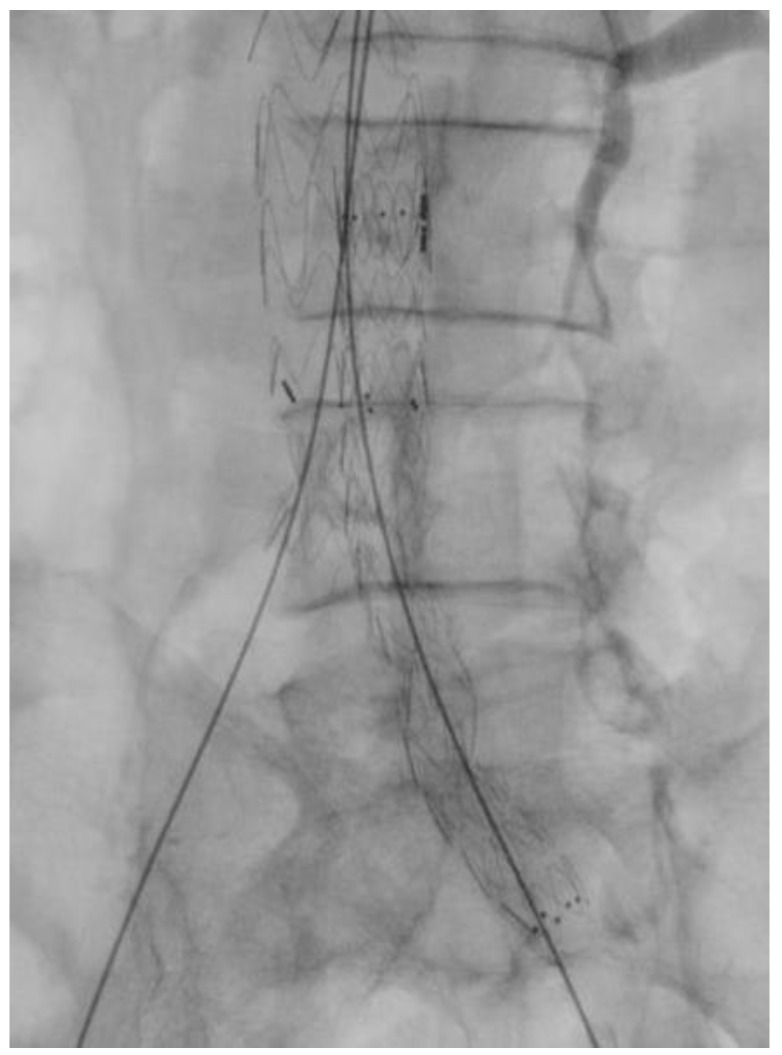
Contralateral limb deployment.

**Figure 3 jcm-13-00175-f003:**
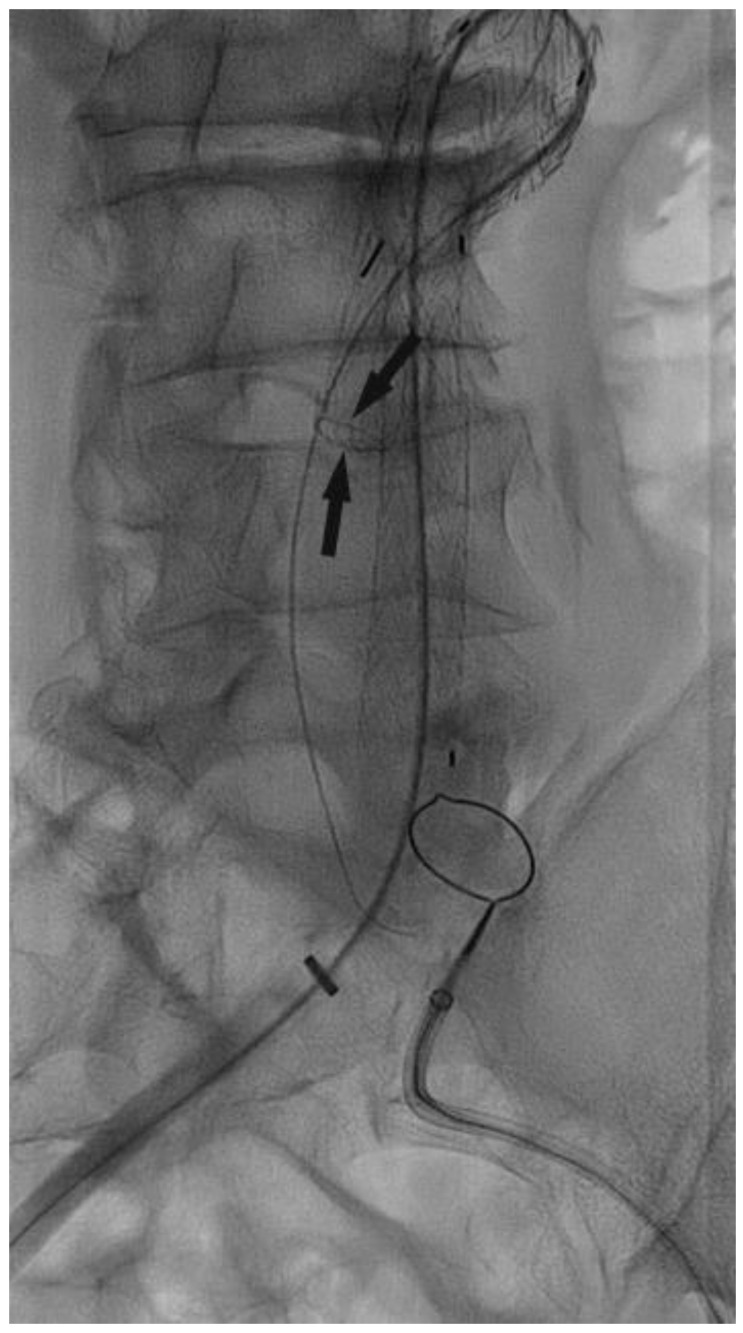
Contralateral gate cannulation. The arrows point to the gate.

**Figure 4 jcm-13-00175-f004:**
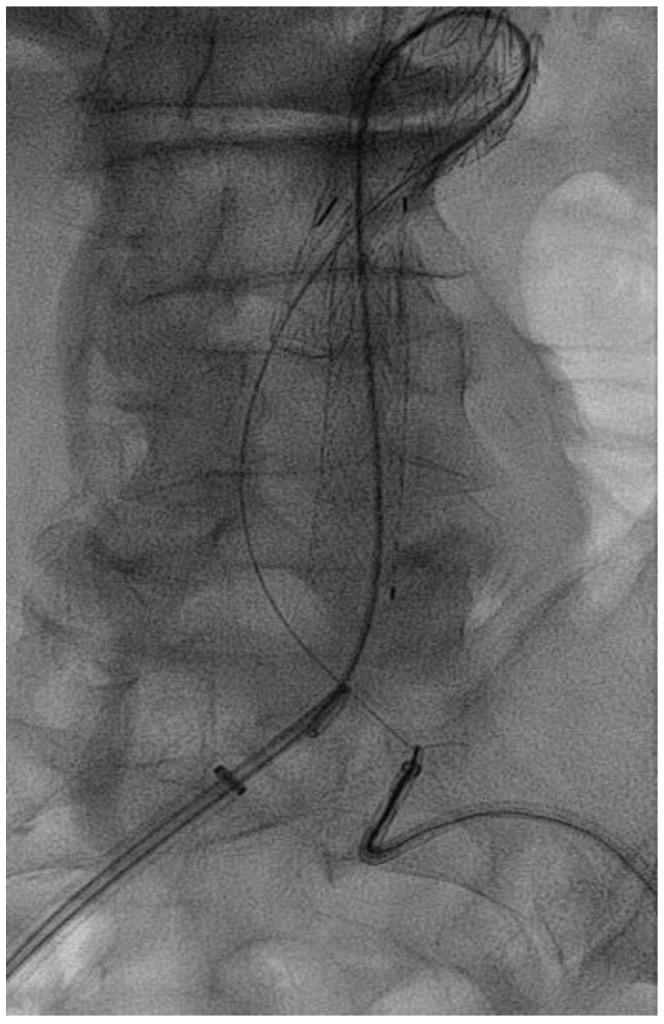
Guidewire captured by a gooseneck catheter.

**Table 1 jcm-13-00175-t001:** Iliac tortuosity index (ratio between the center lumen line distance between the common femoral artery and the aortic bifurcation and the straight-line distance between the common femoral artery and the aortic bifurcation).

Grade	
Absent = 0	<1.25
Mild = 1	1.25–1.5
Moderate = 2	1.5–1.6
Severe = 3	>1.6

**Table 2 jcm-13-00175-t002:** Characteristics of the patients. Continuous data are presented as mean, standard deviation, or median (interquartile range), and categorical data are presented as number (%).

	All Patients (148)	Retrograde (116)	Snare (32)	*p*
Mean Age (y)	73.4	75.7	73.5	0.68
Male (%)	81 (54.7)	62 (53.4)	19 (59.3)	0.67
Neck Length > 10 mm (%)	142 (95.9)	110 (94.8)	32 (100)	0.77
Iliac Tortuosity Index (%)				
0	43 (29.0)	41 (35.3)	2 (6.25%)	0.011
1	42 (28.3)	38 (32.7)	4 (12.5)	0.11
2	40 (27.0)	29 (25)	11 (34.3)	0.37
3	34 (22.9)	18 (15.5)	16 (50)	0.001
Aortic diameter at gate (Cm)	4.49 (3.7–5.7)	4.55 (3–5.5)	4.3 (3.1–5.7)	0.18
Aortic lumen at gate (Cm)	3.07 (2–4.5)	3.07 (2–4.5)	3.13 (2.4–4.2)	0.73
Distance from Bifurcation to gate orifice (Cm)	2.69 (2–4.5)	2.58 (2.7–3.5)	3.14 (2–4.5)	<0.0001
Graft Device (%)				
Endurant II	41 (7,4)	31 (26.7)	10 (32.2)	0.65
Treo	44 (29.7)	35 (30.1)	9 (28.1)	0.87
Zenith	62 (41.8)	50 (43.1)	12 (37.5)	0.68

**Table 3 jcm-13-00175-t003:** Total times of cannulation and for each degree of iliac tortuosity.

Iliac Tortuosity Index	All Patients	Retrograde	Snare	*p*
0	2.76 min.	2.71 min.	3.85 min.	0.0001
1	2.69 min.	2.74 min.	2.8 min.	0.63
2	2.87 min.	3.01 min.	2.71 min.	0.01
3	4.08 min.	4.93 min.	3.28 min.	0.0001
Overall Times	2.98 min.	2.94 min.	3.15 min.	0.33

## Data Availability

The data are not publicly available due to privacy.

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
