# Peer review of "Contralateral Snare Cannulation vs. Retrograde Gate Cannulation during Endovascular Aortic Repair in Difficult Iliac Artery Anatomy: A Single Center Experience"

_jcm, 2023, doi:10.3390/jcm13010175_

Round 1

Reviewer 1 Report

Comments and Suggestions for Authors

This is a paper that presents what is well-known toi practising vascular community that iliac tortuosity can make difficult the retrograde gate cannulsation during EVAR. The authors here present what is expected, the higher the iliac toruosity the more difficult and time consuming the retrograde gate connulation. I would like the authors to clarify the following:

Do they think that in the presence of iliac tortuosity the option of the endograft in a Ballerina mode could help and in which situations?

Do they think that percutaneous access of the left brachial artery is easier than the contralateral snaring tecchnique? 

I understand that hey have not included these techniqjues in their patients - Please clarify.

Nevertheless I would advise them to expand a little their DISCUSSION incorporating these two options

Comments on the Quality of English Language

Quality of English is acceptable. Minor linguistic review may be useful

Author Response

Thanks to the reviewer for the valuable information. We included information on the ballerina technique and percutaneous access to the upper limb in the discussion session. We have adopted this approach on some patients but with unsatisfactory results in terms of success and complications.

Reviewer 2 Report

Comments and Suggestions for Authors

Thank you for conducting this study on contralateral snare versus retrograde gate cannulation during EVAR in a single center. However, there are several critical issues that need to be addressed. I have provided specific comments below to highlight areas of concern and potential improvements:

1.     The study is severely affected by selection bias, a small sample size, and extremely high heterogeneity between the compared groups and subgroups.

2.     The application of the contralateral snare technique only after the failure (>15min) of the common retrograde technique raises a significant concern. This is not a comparison between the two techniques but rather a report of different technical methods of cannulation with a failure to achieve in the second one. This constitutes a fundamental violation of methodology rules for this type of research.

3.     In the Methods section, it is not mentioned whether the cases were consecutive or not.

4.     Table 1, while the recommended grading system regarding ITI is interesting, the criteria for selecting specific intervals between the groups should be clarified.

5.     Table 2, it is evident that the two groups, as well as the subgroups, are not comparable.

6.     Page 4, lines 117-118, it is crucial to specify who performed the control and confirmation of the cannulation. Additionally, was this taken into account during the time measurements? Could this have affected the results in any way?

Page 4, lines 123-124, it's worth noting that brachial access for cannulation of the contralateral gate is commonly considered a last resort and bail-out solution. Other techniques that do not require further access, particularly those utilizing upper extremities like guided catheters or steerable sheaths, should be discussed.

7.     Page 4, lines 134-138, it seems that the two groups are heavily affected by selection bias and are not comparable at all.

8.     Considering all the aforementioned points, the conclusion lacks scientific support from the provided data and cannot be adequately justified based on this study in its current form.

Comments on the Quality of English Language

Needs thorough English editing

Author Response

We appreciate the reviewer's suggestions and will try to respond to each point

1We are aware of the limitations of the study and have specified this

2 the contralateral snare is a rescue method and for this reason it is always preferable to use the traditional method first and not the contralateral snare approach. Being a retrospective study, a direct comparison was therefore not possible.

3 we specified that the cases are consecutive

4 this method was described in the following work that we reported: Chaikof EL, Fillinger MF, Matsumura JS, Rutherford RB, White GH, Blankensteijn JD, Bernhard VM, Harris PL, Kent KC, May J, Veith FJ, Zarins CK. Identifying and grading factors that modify the outcome of endovascular aortic aneurysm repair. J Vasc Surg. 2002 May;35(5):1061-6.

5 Problem we have specified

6 The control of cannulation is taken into account in the measurement time, the use of the contralateral snare is adopted after the use of any device to attempt retrogated cannulation. We also specified that brachial access is a last resort and is fraught with complications

8 Problem highlighted several times and objectively not solvable!!

Reviewer 3 Report

Comments and Suggestions for Authors

clarification about he exact technqiue (which snare used, which catheters used, etc) would be helpful for both instances. Were there differences in time to cannulate in large versus small aneurysms?  between stent graft manufacturers?  More baseline information would be helpful. 

Comments on the Quality of English Language

Minor edits to both the English language as well as formatting would be helpful. 

Author Response

Thanks to the reviewers for the important suggestions. We have specifically described both techniques in the methods session with related images. Unfortunately we have no information on the different cannulation times between large and small aneurysms and between the different endografts.

Round 2

Reviewer 2 Report

Comments and Suggestions for Authors

No further comments.

Comments on the Quality of English Language

Need language editing.

Author Response

There are no further indications from the reviewer. If English needs to be improved, specify which part of the manuscript.